# A Qualitative Exploration of the Needs of Community-Dwelling Patients Living with Moderate Dementia

**DOI:** 10.3390/ijerph18178901

**Published:** 2021-08-24

**Authors:** Tzu-Pei Yeh, Hsing-Chia Chen, Wei-Fen Ma

**Affiliations:** 1School of Nursing, China Medical University, Taichung 406040, Taiwan; tzupeiyeh@mail.cmu.edu.tw; 2Nursing Department, China Medical University Hospital, Taichung 406404, Taiwan; 3Department of Nursing, Tsaotun Psychiatric Center, Ministry of Health and Welfare, Nantou 54249, Taiwan; chia23718662@gmail.com; 4Ph.D. Program for Health Science and Industry, College of Health Care, China Medical University, Taichung 406040, Taiwan; 5School of Nursing, China Medical University Hospital, Taichung 406040, Taiwan

**Keywords:** home-care service, moderate dementia, needs, qualitative study

## Abstract

Few studies have focused on developing a better understanding of the needs of patients with moderate-stage dementia. This study aimed to explore the needs of people living with moderate dementia and receiving home-care services from a local mental hospital. The study adopted a descriptive qualitative approach with purposive sampling to recruit patients with moderate dementia and receiving home-care services. Data were collected by face-to-face interviews and content analysis was used to interpret the experiences in the dialogue data. The results showed that the needs of people living with moderate dementia receiving home-care services contained four themes: the demand for company and care, the wish to recall familiar images, the need of reaffirming life purpose and value through reflection and reminiscence, and the desire for making autonomous end-of-life decisions. In addition to daily care, people living with moderate dementia crave companionship, expect meaningful exchanges of experiences to share their life, and have demands to have a voice in going through the final stage of life. The participants tended to focus more on issues related to the connections between living and dying. The results provide caregivers and home-care service providers with some insights into offering better care for people living with moderate dementia.

## 1. Introduction

With the constant rise in the elder population worldwide, dementia has become a global problem that is addressed by Alzheimer’s Disease International (ADI) [1]. Dementia is defined in the fifth edition of the Diagnostic and Statistical Manual of Mental Disorders (DSM-5) as a neurocognitive disorder [2]; it may be referred to as a group of symptoms that affect patients’ cognition, emotion, and social functions with all aspects of performances beyond normal aging. The prevalence of dementia has reached 6.99% in Asia [3] and was estimated by Taiwan Alzheimer’s Disease Association (TADI) to be increasing at a steady pace in 25% of adults aged over 80 in Taiwan [4]. A new diagnosed case of dementia is confirmed every three seconds, which means that the number of dementia patients around the world will increase by more than triple, up to 152 million, by 2050 [4].

Dementia is a progressive and neurodegenerative disease [2]. It shows large differences in patients with mild and moderate dementia. In clinical settings, two scales are widely used to identify patients with mild or moderate dementia [5]: the Clinical Dementia Rating Scale (CDR) [6] and the Mini-Mental State Examination (MMSE) [7]. The Chinese version of the CDR is broadly used to screen mild, moderate (in score 2), or severe cognitive impairment in community residents [8,9]. The MMSE, on the other hand, is used to assess temporal and spatial orientation, attention and arithmetic abilities, individuals’ immediate and short-term memory, language-related abilities (including reading, writing, naming, comprehension, and operation), and visual-spatial performance; its highest score is 30 [9,10,11]. An MMSE score ranging from 11 to 17 indicates moderate cognitive impairment condition [7].

Dementia leads to the gradual decline of one’s cognitive function, memory, and self-care behaviors [3,12]. Helping patients living with dementia maintain function in everyday activities is seen as a top priority [13]. Caring for people living with dementia requires close attention to their physical and psychological needs and careful observation of their behavioral and psychological symptoms (BPSD) or changes [1,14]. In the meanwhile, anxiety, depression, frustration [15], and loss of hope and meaning in life can cause psychological crises and weaken patients’ physical functions [16]. Nevertheless, a systematic review showed that people living with dementia received insufficient support of satisfying their needs from caregivers or from the health service system [17].

Patients with mild and moderate dementia have different symptoms and cognitive impairment. However, the distinguishability of caring between mild and moderate dementia can be ambiguous. The needs of patients with mild dementia can be found in several studies. For example, maintaining control over one’s life is a crucial need for people with dementia early on [18,19]. Chen and colleagues reported a “tug-of-war” with the passage of time for people with early-stage dementia and indicated their needs for reversing impaired memory and returning independence [20]. Louise, Elizabeth, and Trinity identified that there were two main needs among patients with mild dementia, meeting their self-expectation and maintaining connections with other people [21]. Most of those studies focused on exploring a better understanding of the needs in patients with early-stage dementia. However, patients living with moderate dementia may represent similar needs, which still remain insufficiently understood. People living with moderate dementia may not be able to express their needs the same as those people with early-stage or mild dementia, resulting from their worsening and more severe cognitive impairment and daily function decline [19,20,21].

The difficulty in communication with people living with moderate dementia because of their cognitive impairment makes it much more challenging for researchers to obtain clinical evidence with moderate dementia patients’ expression of their needs. Trying to understand and respond to their needs is crucial to the development and implementation of better care for dementia patients in later stages. This study aimed to explore the firsthand experiences of people living with moderate dementia to understand their needs in depth, and to make better care plans according to the research results and improve the quality of home-care services.

## 2. Materials and Methods

### 2.1. Research Design and Participants

The study adopted a qualitative research design with purposive sampling to enroll adults living with moderate dementia who were taken care of by families at home and receiving home-care services from a local mental hospital in Taiwan. A potential participants list was obtained from a home-visit nurse team from a mental specialist hospital. Nurses who were familiar with the patients and families under a long-term caring relationship phoned the families to enquire whether they and the patients were willing and able to undergo an interview. Oral consent was obtained via phone communication and formal informed consent forms were obtained when the researcher visited together with home-care nurses. While the interview was conducted, the home-care nurses and families accompanied the patient and researcher in order to ease the patient’s potential anxiety, worry, or sudden outburst of emotion; the nurse and families helped in clarifying unclear sentences or in interpreting patients’ presenting meanings. Semi-structured in-depth interviews were conducted face-to-face to collect data. The inclusion criteria were (1) dementia diagnosed by psychiatrist or neurologist; (2) CDR score of 2 (representing community residents with moderate dementia); (3) MMSE score ranging from ~11 to ~17; (4) capability of verbal communication; and (5) willing to sign an informed consent from the participants and their primary caregivers. The exclusion criteria were (1) premorbid IQ < 70; (2) current abuse of alcohol or drug, or more than a 5 year history of alcohol or drug abuse. The interview was terminated when they (1) became agitated or provided other unexpected responses during the interview; (2) families or home-visit nurses thought the interview should stop. When analyzing the data, including seven participants, no new category could be found; in addition, half a year later, the research team no longer found any home-care moderate dementia patients who were able to communicate clearly. Due to the time limitation, this study terminated after recruiting seven participants.

### 2.2. Data Collection

The interviews were performed based on the principles of accessibility, privacy, and comfort. The interview venue was set at the participants’ home with their caregivers considering the convenience and sense of security for such special participants. Prior to the interview, the researcher explained the study purpose and the procedures to both the interviewees and their caregivers. An interview, lasting 20–30 min, was recorded and later transcribed into a verbatim transcription. Each participant received 2–4 interview sections. Notes were made while interviews were conducted and were subsequently identified as “reflection journals” for analysis. Six guiding questions of the semi-structured interview are listed below, and probing questions were used to change sentences or questions so that they were easier to understand for participants; sometimes, the caregivers helped in altering the question for better understanding. The semi-structured interview questions were validated by three researchers with cautious concern given about the participants’ ability to understand and answer:How are you feeling today?Would you like to share with me how you have been feeling recently?What life experiences would you like to share?Do you have any needs/wishes that you want me to know?Would you mind providing some examples to describe your needs or wishes?What are the most important things that you care about at this moment and why?

The interviews were completed by an experienced psychiatric nurse with more than 20 years of clinical experience, having experience in home-care visit services for more than five years and in dementia care for more than 10 years. The interviewer had received qualitative research training and conducted qualitative research. The researcher applied some strategies to help the participants express themselves more, such as active listening, encouraging attitude, no judgement, prolonging the time for thinking and answering, and involving the caregivers to help in rephrasing questions.

### 2.3. Data Analysis

All interviews were transcribed verbatim within 48 h after the interview in order to facilitate qualitative content analysis, which followed the steps suggested by Berg [22], Rubin, and Rubin [23], and Strauss and Corbin [24]. Open coding was applied after data were transcribed into text, and the reflection journal was carefully reviewed several times. The codes were classified, and those with similar contents were grouped into one category. Categories and subcategories were then developed and themes and subthemes were established. The software Nvivo 12.4^+^ was used to manage data during the data analysis process. In some circumstances, the codes with transcriptions were difficult to interpret, and the researchers had to listen to the digital recording again and look into the scenarios; they then took the reflection diary into account for further discussion in data interpretation.

### 2.4. Study Rigor

The rigor in this qualitative research was guided by Lincoln and Guba for reaching the credibility, transferability, dependability, and conformability of the study [25], and strategies were guided by Korstjens and Moser [26]. The research team members ensured the accuracy of the verbatim transcription, and double-confirmed the code and categories formed in the analysis processes. Meanwhile, each participant received 2–4 interview sections with an appropriate rest time or any activity they requested to avoid confusion or incomprehensive communication caused by fatigue. This prolonged engagement and the several short interviews with observation were to increase the confidence of the truth of the research findings. The data involved the conventions, behaviors, experiences, and the context, including participants, family caregivers, and the environment. Those contexts were analyzed to make sure the behavior and experiences became meaningful to an outsider to increase the transferability.

Simple replies such as “yes” or “no” were asked when the researcher wanted to confirm the participants’ understanding of questions or the researcher’s interpretation in the meanings of answers. A worksheet was used to trace the work progression, and a reflection diary helped the researcher to recall the context in the interview and ask other research members to clarify when a dilemma occurred in coding, emerging categories, sub-themes, and themes. The complete process from initial coding to final emerged themes was checked by a professional in dementia care and qualitative research. Reflexivity was checked by the memos and research team discussion. Those strategies were to increase the dependability and confirmability.

### 2.5. Ethical Considerations

The study was approved by the institutional review board of the study site (IRB number: 104002). Prior to conducting the interview, the participants and their families were briefed on the study purposes and the interview procedures for obtaining an adequate understanding of the study before signing the informed consent. Throughout the study, the feelings, viewpoints, and experiences expressed by the participants were collected, understood, and analyzed with a respectful and nonjudgmental attitude. Moreover, the names of the participants were coded for privacy protection, and all the interview data were used exclusively for academic purposes.

## 3. Results

### 3.1. The Participants’ Information

Among participants living with moderate dementia, four were male and three were female. The participants were aged between 68 and 87 years, and the mean age was 78.4 years. The demographic data are listed in Table 1. On the inclusion criteria evaluation, all participants scored 2 in CDR, and the MMSE ranged from 16 to 17 with an average of 16.71, which meant all of them had moderate dementia.

### 3.2. The Needs of the Participants Living with Moderate Dementia

The needs of the people living with moderate dementia focused on company, keeping memory, finding the meanings in life, and preparing for death. Four main themes emerged, and the nine sub-themes are summarized in Table 2.

### 3.3. The Demand for Being Company and Cared

Participants expressed a need for both care and companionship, as indicated by the two subthemes.

#### 3.3.1. Need to Be Listened to

Participants were quite upset about not being able to do things that were once their pride. They needed someone to listen to their frustrations about not being able to perform familiar activities. In spite of being diagnosed with moderate dementia, the participants appeared to perceive no severe functional decline and tried their best to not rely on others. However, gradually losing dependence kept them away from engaging in activities they usually performed by their own. Participants were quite upset about not being able to do things they were once good at, and they hoped to have someone listen to them about the limitations and worries of not being able to complete proficient activities anymore. For example, the kitchen was no longer their territory, and riding or driving became a totally restricted task. Sometimes, the participants mentioned some tasks they still can do excitedly.

Participant 01 proudly told the interviewer: “*I know how to do my laundry by using a washing machine…I kept the machine running from evening to dawn. I took out the clothes to dry when I saw it*
*in the*
*morning*.” However, she was angry about her inability to cook: “*Where are those pots and pans I bought? I can’t even cook for myself now,*” she felt quite upset about not being able to do something just as she did in the past in daily life. 

Similarly, participant 04 said: “*I want to work on the farm and drive around to inspect the fields. There are so many things I want to do. But I am no longer able and allowed to do those things now*!*”*

Participant 07 expressed the same feelings of helplessness with upset: “*They don’t let me drive…or ride. They took away the keys of my car and motor scooter*.”

#### 3.3.2. Need for Company

Participants felt a strong need of companionship with someone who is understanding and caring for their lives and needs. When their memory was no longer reliable, and communication became incomprehensible, the participants’ desires for company were expressed. In addition, someone who knew the patients’ past well were needed to ease their agitation, anger, or anxious emotions.

In terms of the company needed from the interviewer, participant 03 said: “…*Walking like this, I mean with someone besides me, makes me feel much better!*” During the walk, the old man noticed on the ground a betel nut box showing a picture of a girl in a bikini. He bent down to pick up the box: “*Let’s take a good look at her (laughing). Isn’t she beautiful?*” He happily placed the box in his shirt pocket. In this scenario, the participant showed the need to have company and feel comfort and relax when someone was walking together with him.

When asked what he cared about the most at the moment, participant 04 turned to his wife, who answered the question for him with a smile: “*For him, it’s definitely not being able to work on his rice farm field!*” To the interviewer’s next question “*Anything else?*” He again turned to his wife for asking help, and she told the interviewer: “*He is also complaining all the time about no longer being able to ride around with his motorcycle!*” Participant 04 kept nodding and smiling. In this scenario, the participant showed the need of being represented by a close family due to decline of his presentation and communication skills. Therefore, the need of company was apparent. 

Participant 06 was an old man (86 years) who had not seen his mother in China for nearly 40 years after he retreated with the government from China to Taiwan in 1949. By the time the ban on Taiwan-China cross-strait travel was lifted, he was hurried back to China but faced the tragic fact that his mother already passed away. After being diagnosed with moderate dementia, the old man frequently reported episodes of suddenly taking leave of his family to travel on foot back home to his mother. According to the family’s description, his wife (who died two months ago) would walk with him, comforting and coaxing him until he grew calm and sensible enough to return home. For one of the interviews with participant 06 scheduled at his home, the interviewer witnessed the old man pacing back and forth in agitation, demanding to have his late wife accompany him to embark on another home-returning journey to his mother in China. He was too agitated to listen to anyone until the interviewer, assuming the role of his late wife, walked him out of the house. With the company of the interviewer, the old man started to moan: *“The darkening sky looks as if it were dying in an afternoon at summer’s end. It’s so depressing…Why am I here? I simply have no idea what I’m doing here...Now I remember. I’m going home.”* With his finger pointing at the sky, the old man continued, *“Going back home over there. Two more mountains and 40 miles, then I’ll be home, and everything will be alright once I’m home.”* From the researcher’s observation, participants need a great quantity of time for company.

When interviewed at her home, participant 01, a 79-year-old lady, suddenly cried out *“My mother can’t get out of her bed! I need to help her in hurry. She’s dying!”* Failing to remember that her mother had passed away a long time ago, the old lady seemed to be worrying greatly about her mother’s health. Soon after those words, she became even more anxious. The daughter of participant 01 walked to the window yelling *“Mom, Mom, you need to get here quick. The dogs are trampling your vegetable garden!”* Then, participant 01 stopped crying and said *“What! I hear you. Hurry up! Come with me to get rid of those dogs.”* It took quite a while for her family members to deal with her unstable moods.

### 3.4. The Wish to Recall Familiar Images

The progressive symptoms of memory loss and cognition impairment made participants with moderate dementia hope to be capable of recollection; the recalling process may sometimes help to divert their attention. The second theme incorporated two subthemes related to the continuous impacts of deteriorating cognitive function on their emotional wellbeing.

#### 3.4.1. Over-Increasing Severity of Forgetfulness and Confusion

Participants might still find people and things around them familiar but continue to experience difficulties in naming or recognizing them. Participants became increasingly incapable of recognizing the people and surroundings with which they were familiar.

When asked about her age, participant 02 smiled and said: “*I am…I don’t know for sure. 70, I guess. Born in the Year of the Dog. I need someone to accompany me in my remaining life.”* However, the age that the participant claimed was wrong. Her true age was 80 years. Born in the Chinese Year of the Dog is correct.

Participant 03 tried hard to answer the same question but ended up confessing “*I have no idea how old I am, but I know I am old, very old*.” When the interviewer requested to introduce his wife and son sitting next to him, the old man could not recall their names and said: “*She is one of my elders, but I can’t tell which one.*” Then pointing at his son, he said: “*This is my brother.*” His wife burst into laughter upon hearing these words. 

Participant 07 said quietly: “*I…soon… I’ll no longer dare to drive on the Street*.” The interviewer asked him what happened and he replied: “They said *I got lost easily when driving on the street…That’s why I live in Taichung City* (in central Taiwan) *now!*” When the interviewer checked with his daughter, she said that none of the things her father mentioned had happened.

Participant 06 complained about often forgetting to have his meals and taking his medication: *“I can feel an obvious decline in my memory.”*

#### 3.4.2. Tendency to Deal with Anxiety and Frustration with Evasion

When feeling confused or embarrassed during the interview, most of the participants chose to flee from the scene, instead of asking for a break. With cognitive impairment and memory loss, those familiar people and things may turn into fragmentary or even unrecognizable pictures, and the participants could not do anything but just feel increasingly frustrated, anxious, and insecure. Trying very hard but still failing to identify someone or something often prompted the participants to flee from the venue of the interview.

Participant 03 was unable to tell his own age and name family members and grumbled impatiently: “*Stop it. I don’t want to listen to this! I need to go out for a walk.*” Participant 05 had recently been troubled by dream–reality confusion and visual hallucinations, claiming “*My land has been bartered away recently and someone has been stealing my property. I need help safeguarding my fortune.*” Deliberately lowering her voice, participant 05 asked the interviewer not to discuss their conversations with the nurse hired to take care of her at home: “*Don’t tell her anything so she has nothing to tell people…*” The interviewer inquired several times into the reason of her worry, but she just kept looking at the interviewer in silence and confusion.

When family members complained during an interview about participant 07 causing constant troubles to his sons, instead of responding to the complaint, participant 07 whispered to the interviewer: *“I have serious suspicion that people are stealing from me, but my memory…it’s getting worse and worse, and I have become lazier and lazier for walking. There is nothing I can do about it.”* With these words, the old man fled from the interview right away and went into his room. He showed poor attitude to his families on blaming him and restricting his activity. In addition, he felt embarrassed and bewildered. The participant did not agree with his family’s point of view and protested for himself by leaving the scene.

When his son frowned with the whisper, “*My father’s condition is deteriorating,”* participant 04 stood up in an attempt to leave, saying, *”I am feeling dizzy.*”

### 3.5. The Need of Reaffirming Life Purpose and Value through Reflection and Reminiscence

The third theme incorporated three subthemes that concerned the need of regaining self-worth by reviewing past experiences.

#### 3.5.1. Increasing Difficulty in Recognizing Self-Worth

Recognizing self-worth became increasingly difficult because of the progressive cognitive and functional decline. With the growing inability to perform daily activities, independency appeared to generate an urgent need of the participants to seek self-recognition. The task, however, became more and more difficult because of the progressive deterioration in their condition.

Participant 01 described herself in a negative point of view: “*I’m a nuisance to everyone. Nobody wants to do anything with me! See those caring and filial daughters-in-law of my friends, I thought to myself ‘When will I be blessed with such a daughter-in-law?’ How was I supposed to know that my daughter-in-law made my life miserable*?” After venting her feelings, she added: “*I am not a picky mother-in-law. I’d be happy as long as there’s food for me*,“ thus revealing a vicious circle of changing moods.

Participant 03 experienced a fall from his bike a couple of days before the interview. When his son asked him not to ride a bike on the street under the scorching sun at noon, he lashed out loud with his disapproval: *“Are you kidding with me? That’s a wide road, and I used to ride on it all the time. How come I’m not allowed to do this now? It’s unfair! Who are you? How can you look down on me like that?”* Turning to the interviewer, the old man continued: *“They are a bunch of nitpickers, always nagging me. It’s ridiculous!”*

#### 3.5.2. Fondness of Reminiscing about Past Significant Events

Participants searched for the meanings and values of their lives through reflections on past important events or happy memories. The participants expressed their inner feelings and traced how they underwent and survived their changing roles throughout their life. Due to their severely declining short-term memory, the participants talked about their past and mentioned important or impressive meaningful events a lot. This process may be regarded as a major method of self-recognition or self-assurance in this population.

Participant 05 reviewed her life: “*Childhood was a happier time for me. My father*
*couldn’t bear to see us little children toil on the farmland. He would send us home, saying that the weather was getting too cold…I was loved before I got married when my life took a bitter turn. Whenever my mother-in-law and I were together, like we are now (participant and interviewer), I was in constant fear that I might say something that offended her. Any slight slip of tongue from me was met with an outburst of anger and a threat to leave home…Nothing much has changed now that I become a mother-in-law myself. Still, I have to worry all the time about being scolded by my daughter-in-law, who is bent on being mean and hostile to me. How I envy those with a decent daughter-in-law!”*

Participant 02 mentioned about her marriage too: “*A matchmaker introduced him to my family. My parents found him likable, and I became his wife.”*

With regard to his childhood, participant 04 said: “*We lived in Tucheng* (a small township in northern Taiwan) *for four years when I was a child…We**,*
*children**,*
*helped our father build a direct track across the fields to shorten our trip to school. We were happy, and so were our neighbors.”*

Participant 06 remembered: “*After getting married, I worked all day on the farmland and went home at dusk. My wife and I rose with the sun. She was cooking, and I was getting prepared for another day’s work on the farm…”*

#### 3.5.3. Reliance on Compliance with Moral and Social Norms to Build Self-Recognition

Participants found comfort and value in life by celebrating compliance with moral and social norms. When reconstructing past events and experiences through reminiscence, the participants were also reaffirming the moral and social norms guiding their lives. Whether they and their children complied with or deviated from those norms had left deep imprints on their minds. When the participants perceived the respect and filial obedience from their children or young generation, they felt with dignity, value, and affirmation.

Participant 02 spoke highly of her filially pious sons and grandchildren but remained wary when it came to the distribution of his legacy: “*My sons never gossip or bug me with my money matters. My eldest son in particular does not complain about who gets what and how much... I wouldn’t know who*
*I should listen to if they started arguing over the distribution of my money and property after I pass away. That’s something I don’t do as long as I’m still alive.”*

Participant 04 also found comfort in his having filially pious sons and grandchildren, and his wife sitting next to him added: “*And our daughters-in-law too. They have all been very sweet and thoughtful.”*

### 3.6. The Desire for Making Autonomous End-of-Life Decisions

The fourth main theme covered two subthemes involving control over end-of-life issues.

#### 3.6.1. Need for Religious Faith-Based Strength

Religious faith remained rooted in participants as an ultimate source of guidance, protection, love, and understanding. At their age and with their conditions, most of the participants had passed down the traditional religious rituals to the younger generations in their families. Religious faith, however, remained an important role in family inheritance. Religious faith supports the patients to continue their life in irreplaceable strong belief in their minds. The patients showed their needs of being blessed and protected from God in health, safety, finance, and even whole family. 

Participant 01 said solemnly: “*God listens to all prayers that come from my heart. One day, the sky was dark, and I felt cold. So I clasped my palms together to pray: ‘My dear Lord in heaven, I am feeling cold. Please have the sky turned clear.’ In a moment, the sun came out.”* At this juncture, her daughter told the interviewer that the participant had experienced deteriorating memory, forgetting her family members one by one. In response to these words, participant 01 held the interviewer’s hands and said with great emphasis: “*In*
*times of difficulty and desperation, I never forget to lift my hearts and hands to God in heaven because I truly believe that our Lord is a God of righteousness and justice. It’s fine with me if I forget anyone and everyone, but I shall never forget my Lord!”* Participant 02 smiled and said: *“The Lord of Mountain is the patron of our six small villages.*
*Joining His annual grand patrol and pilgrimage brings you and your family good luck and good health. Taking part in religious rituals is always beneficial**.”*

Participant 05 told the interviewer: “*If you make any vow while praying to gods, you must remember to redeem the vow. What a pity that, after the surgery, I can no longer go upstairs to pay my tribute to gods.”*

Participant 04 believed in Taoist: “*No need to complain to gods as they know, in their omniscient way, everything that’s weighing on your heart. Honor gods with your awe and reverence and you shall be blessed. Anything favored by gods is bound to be a success.”*

#### 3.6.2. Preparedness to Face Death on Their Own Wishes

Participants appeared to be prepared in facing death in their own ways. Due to knowing well that they were approaching the end of their lives, the participants craved to remain capable of controlling the process of dying and how to deal with death when the time came.

When choking on water during an interview, participant 06 tried hard to find the right words for the episode: *“What was that? Was I…was I choking?”* The old man then grew frightened, asking *“Am I going to die?”* The interviewer patted his hands to comfort him, and he said: *“I don’t want to die from choking. I’d like to die in my sleep!”* The interviewer raised another question, asking the old man whether he believed in Buddha or in God (in Christian). *“God. I believe in God,”* participant 06 answered emphatically. The interviewer smiled and told him: *“Then, when you are ready to go, be sure to follow His angels to Heaven. Okay?”* The old man nodded in all sincerity.

*“I was quite a different person when I was young,”* said participant 03. *“After they told me that I was sick, I went to every temple I knew, burning incense and praying to god on bended knees. I went through an array of rituals, and I donated profusely, seeking for god’s blessing that I could get rid of the illness. Now, I’ve learned to let go. When you let go of your obsessions in life, your mind becomes open and peaceful. Cars, houses…all those things you used to care so much become so trifling to merit your concerns.”* Twenty minutes after this rare moment of insight, the old man forgot what he had said and whom he was talking to. Soon, the interview ended with a family member holding his hand and taking him out for a walk.

Upon seeing the interviewer showing up for a scheduled interview, participant 04 burst out: *“I’m 78 years old, and I’m dying! Are you here to snatch my soul away?”* This showed the fear of facing death.

After uttering a few names, participant 07 told the interviewer: *“They were my good friends. All dead now. One died of cholecystitis, rather unanticipated. I missed them a lot…I hope they’ll bury me next to my wife.”* When the interviewer held the old man’s hand and told him that he was a good father, tears welled up in his eyes. The interviewer continued to respond to the participant: *“Look at your daughter. She flew all the way back to Taiwan just for you. And your son too, visiting you all the time. Only a good father deserves that. You’re a good father. Don’t forget that!”* The participant then said with great seriousness: *“When I am ready to go, I’ll remember to follow the light with all my heart.”*

## 4. Discussion

This study recruited seven participants living with moderate dementia. This qualitative study explored the experiences in-depth of their needs from the firsthand face-to-face interviews, which went through difficult processes during research. Four main themes were extracted to share the needs of people with moderate dementia, including the needs of company and caring for lives, the chance to recall memory, the evidence to identify themselves, and the capability of making the final decision for issues with their death. Based on these outcomes, participants showed needs of being cared and feeling safety a sense of safety or security, self-identification, and autonomy. Regardless of the participants realization of the large impact by the disease, those needs revealed that participants wanted to be seen as a person with a worthy past, to maintain a stable life status, and to have courage to face their remaining life.

From the first theme (demand for company and being cared for) and second theme (wish to recall memory), participants showed their strong feelings of frustration and insecurity. The main reason may be the behavioral and psychological symptoms of dementia becoming more serious [27,28] in the moderate stage of dementia than the early stage. Therefore, it is more difficult to complete the daily routines by themselves, especially dealing with daily activities demanding attention and detailed information [29]. Other studies have also reported similar results, for instance, Erkkinen, Kim, and Geschwind reported that the daily life of patients with Alzheimer’s disease was affected by impaired recent memory, visual space, and executive function [30]. They often misplaced items, repeated their conversations, or repeatedly asked the same questions, and they could not remember dates or appointment times; eventually, it became difficult to maintain personal life [13]. Those gradual impairments led to disability, feelings of insecurity, and an increasing loss of independence. Participants felt they would be better if there was someone who knew them well and was willing to take care of them. This may differ from the people living with early-stage dementia, who desire to be more independent by themselves. Thus, we may conclude that even patients living with moderate dementia still desire to keep part of their autonomy and being esteemed.

Most patients living with dementia generally suffer from emotional distress, including fear or anxiety [31,32], and depression [33]. Similar results were found in this study. The participants showed a gradual breakdown of cognitive abilities, and this distorted cognition continued to affect their mood. This situation made the family constantly need to bring familiar images back in mind again to diminish their fear of forgetfulness or dealing with the unstable emotional states. With the progression of disease, patients living with dementia have significantly increased day-by-day physical, mental, social, and spiritual care needs [34]. From themes one and two, the results suggested that patients living with moderate dementia need to be accompanied by someone who was familiar with them. Moreover, in terms of dementia care, it would be easier to turn their attention away when dealing the patients’ behavioral and psychological symptoms. Meanwhile, providing familiar images surrounding their environment may help to comfort patients’ emotions and increase their sense of security.

The third theme of this study showed that participants were experiencing a self-redefinition process. This process seemed to require one’s own past experiences and memory, combined with the unique value and the social culture context to complete. These findings were seldom discussed for people living with a moderate stage of dementia due to their constant change and complex symptoms. Górska et al. mentioned that people living with mild dementia crave being connected with memory [35]. Chen et al. also showed that patients living with mild dementia struggled to find a balance between independence and dependence in the process to re-establish their sense of self-worth [20]. Therefore, this study found that regardless of the severity of dementia increase, every individual still had a strong need to re-define self with certain values. This may be related to the integrity versus despair of completing life tasks in the old age period [36,37]. However, self-integration may particularly be difficult for those people living with dementia. Daly et al.’s research pointed out that most of the patients living with dementia believe that the continuity of connecting the past, the present, and the future is quite meaningful to them [17]. As a result, it may have a great effect on the comfort of patients when they face difficulties in self-recognition if caregivers can often listen to them talk about past glorious experiences and wonderful memories.

The fourth theme is related to the need for autonomy in their remaining life. They emphasized the importance of spiritual support to help them in achieving autonomy in facing the end stage of life. In this study, it is believed that spirituality and religious faith can be one of the useful methods for patients living with moderate dementia to cope with their chronic health conditions [38]. Daly et al. and Chen et al. even proposed that spirituality or religion is an important source of strength in facing the challenges of progressive cognitive impairment for patients living with dementia [17,20]. The spiritual power also provides them with courage and meaning in life to allow the patients to be able to respond to the loss of memory and cognitive function [39]; spirituality may assist them in finding and maintaining hope, and also discover the meanings of life [17,27]. This study echoes that spiritual support is helpful in dealing with the needs of self-identification and autonomy for patients living with moderate dementia.

The study explored the needs of people living with moderate dementia and receiving home care. The data saturation was not solidly confirmed, because of the time limitation in finding appropriate participants. Sometimes, the interview would be interrupted several times due to the participants’ sudden confusion or difficulties in expression. Although the complex communication and constant change in the participants’ situation made this study very challenging, this is a rare opportunity to realize moderate dementia patients’ needs and thoughts. The major limitation in this study is that the results may not be applicable to adults living alone or allocated in a nursing center. In addition, the participants in this study were all from Taiwan and the study outcomes as references should be limited to patients living in other culture context areas. The research results may not be generalized to people living with mild or severe dementia.

## 5. Conclusions

This study explored the needs of participants living with moderate-stage dementia, and they made great efforts to retain life balance and self-identification in their remaining life. Four main themes emerged to provide an in-depth understanding of the needs in adults with moderate-stage dementia. In addition to daily care, people with moderate dementia crave companionship, expect meaningful sharing with their life experiences, and demand to have a voice in steering the process in the final stage of life. The needs expressed by the participants may provide caregivers, families, and home-care service providers with insights into offering better care to people living with moderate dementia by properly responding to their needs and wishes by additionally concerning and evaluating the needs of esteem and spiritual support. The strategies of palliative care in preparing death, such as saying sorry, saying thanks, saying love, and saying goodbye may, be introduced in this field in the future.

## Figures and Tables

**Table 1 ijerph-18-08901-t001:** Demographic data of participants.

Participant	Gender	Age	Education	CDR	MMSE	Religion	Notes
01	Female	79	Illiterate	2	17	Christian	Diabetes after brain tumor operation; vascular dementia
02	Female	80	Illiterate	2	16	Taoist	Operation for prolapse of uterus; Alzheimer’s disease
03	Male	87	Middle school	2	17	Taoist	History of myocardial infarction; Alzheimer’s disease
04	Male	78	Primary school	2	16	Buddhist	Stroke; vascular dementia
05	Female	71	Illiterate	2	17	Christian	Hypertension; diabetes mellitus; mixed vascular dementia
06	Male	86	Primary school	2	16	Christian	Scoliosis corrective surgery; vascular dementia
07	Male	68	High school	2	17	Taoist	Degenerative arthritis; 2/3 gastrectomy; dementia with Lewy bodies

**Table 2 ijerph-18-08901-t002:** The needs of participants living with moderate dementia.

Main Themes	Subthemes
1. The demand for being company and care	Need someone to listen
	Need for companionship
2. The wish to recall familiar images	Worry over increasing severity of forgetfulness and confusion
	Tendency to deal with anxiety and frustration with evasion
3. The need of reaffirming life purpose and value through reflection and reminiscence	Increasing difficulty in recognizing self-worth
	Fondness of reminiscing about past significant events
	Reliance on compliance with moral and social norms to build self-recognition
4. The desire for making autonomous end-of-life decisions	Need for religious faith-based strength
	Preparedness to face death on their own wishes

## Data Availability

These study data are identified participant data. The data that support the findings of this study are available beginning 12 months and ending 36 months following the article publication from the corresponding author, W-FM, upon reasonable request at lhdaisy@mail.cmu.edu.tw.

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
