# Peer review of "A Qualitative Exploration of the Needs of Community-Dwelling Patients Living with Moderate Dementia"

_ijerph, 2021, doi:10.3390/ijerph18178901_

Round 1

Reviewer 1 Report

The authors have tried to answer a relevant research question. However, the study lacks methodological rigor of a qualitative approach and its analysis

Abstract

  1. It is unclear if authors are referring to patients received home care services (from a health facility) or home-based acre

Introduction –

  1. The literature review is not comprehensive. The introduction needs more information on the existing evidence of needs of people with dementia in similar contexts.
  2. Methodology –The methodology section lacks adequate details to assess the methodological rigor of the study.
    1. Was the sample size enough? How did the authors decide on the number 7? Was data saturation achieved?
    2. What was the setting in which study was conducted? Geographical location? Which country? Participant profile?
    3. Who conducted the interview? What is the researcher profile?
    4. Where probing questions used?

Results

  1. Quotes have not been correctly mapped to themes. Also, most of the quotes and experiences described relates to the anxiety and psychological distress that patients’ experiences. They are the experiences. How did the authors map them directly to needs?
  2. Examples

Need for companionship-

He bent down to pick up the box: “Let's take a good look at her (laughing). Isn’t she beautiful?” He happily placed the box in his shirt pocket- This quite does not reflect need for companionship

Need someone to listen

Participant 01 proudly told the interviewer: “I know how to do my laundry using a wash-153 ing machine…I kept the machine running from evening to dawn. I took out the clothes to dry when 154 I saw it in the morning.”

  1. Some of the quotes are from the carers of the patients. SO were these interviews with patients and their carers – authors need to mention this in detail in the methodology

  1. In table 4, some of the subthemes are not needs, but are issues faced  Worry over increasingly severe forgetfulness and con-fusion

  1. Needs extensive English language editing

Author Response

To the Reviewer:

Thank you very much for taking time from your busy schedule to comment on our manuscript. Our responses to your suggestions and questions are summarized below, with reference to the appropriate pages in the text. Revisions in the text have also been highlighted in RED. We truly appreciate your thoughtful and constructive comments that help make this a better paper. We are very grateful to you for all the comments and insightful suggestions that help enhance both the quality and readability of our paper. Thank you.

  1. The authors have tried to answer a relevant research question. However, the study lacks methodological rigor of a qualitative approach and its analysis

 Ans: Thanks for reviewer’s comment. The authors agree and added new 2.4 Study Rigor section in this revision. We added more details in the methodology part. We provided the information of what steps and strategies this study used for reaching the rigor, especially in data collection and data analysis. We hope new 2.4 section in this revision making the study rigor clearer. Please see page 4, line 155-176.

  1. Abstract- It is unclear if authors are referring to patients received home care services (from a health facility) or home-based acre (receiving home care services from a mental hospital)

 Ans: Thanks for reviewer’s questions. In this study, the participants came from the people with moderate dementia in Taiwan live at home and are taken care of by their families, also receiving home care services from a mental hospital. We added some description in the text. Please see abstract in page 1, line 21 and page 2, line 93-96.

  1. Introduction –The literature review is not comprehensive. The introduction needs more information on the existing evidence of needs of people with dementia in similar contexts.

 Ans: Thanks for reviewer’s comment. The authors have revised the literature review section in order to offer more updated evidence. However, most studies focused on the needs of patients with early stage of dementia, not for patients with moderate dementia. That’s why this study tried to explore the firsthand experiences of people living with moderate dementia to understand their needs in depth by using qualitative approach. We have five paragraphs in the Introduction section. High prevalence of dementia in the first paragraph to show the important of this issue and different stages of dementia in the second paragraph. The third paragraph is the general symptoms of dementia and fourth is about needs of patients with early stage of dementia only. The final paragraph showed seldom research on needs of patients with moderate dementia and the important of this study aim. Please see page 2, line 69-90.

  1. Methodology –The methodology section lacks adequate details to assess the methodological rigor of the study.

 Ans: Thanks for reviewer’s comment. The authors tried to describe more details of what we have done in data collection and analysis by perform the process in steps and strategies. We added new 2.4 Study Rigor section in this revision. We provided the information of what steps and strategies this study used for reaching the rigor, especially in data collection and data analysis. Please see page 4, line 155-176.

  1. Was the sample size enough? How did the authors decide on the number 7? Was data saturation achieved?

 Ans: Thanks for reviewer’s questions. The author agrees that sample size is an important study issue. We had notice that when five participants were interviewed and the data was analyzed, there was no new category could be formed. Then, two more participants were recruited, still, no new category was found in data. After half year pass, it is difficult to find appropriate participants with moderate dementia and were able to communicate with their family company. Due to the time limitation, although we were not sure the data saturation was reached, we ended this research and describe this limitation in discussion. Please see page 12, line 527-534.

  1. What was the setting in which study was conducted? Geographical location? Which country? Participant profile?

 Ans: Thanks for reviewer’s questions. The participants and their families were introduced by home visit nurses in a mental hospital, under oral consent in telephone, the researcher visit the participants at home with the company of their families and the home visit nurse in appropriate time. A new demographic table was added in the context to show the geographical location and participant profile. Please see page 4, table 1.

  1. Who conducted the interview? What is the researcher profile?

 Ans: The researcher possesses psychiatric nursing experiences more than 20 years and have five years working experience in home visit care, in addition, the researcher have cared for dementia patients more than 10 years. All researchers received qualitative research trainings and have completed qualitative research before. We added this information in the revised manuscript. Please see page 3, line 136-142.

  1. Where probing questions used?

 Ans: Yes, except six guiding questions, probing and simple questions were used. This information was put in the manuscript. Thanks for reviewer’s this question. Please see page 3, line 129-135 and page 4, line 168-170.

  1. Results-Quotes have not been correctly mapped to themes. Also, most of the quotes and experiences described relates to the anxiety and psychological distress that patients’ experiences. They are the experiences. How did the authors map them directly to needs?

 Ans: Thanks for reviewer’s questions. The authors agree this is significant but difficult tasks in this study. We followed the steps of content analysis and analyzed the data according to the reflection dairy and recall the scenarios of interviews, also from the researchers’ experiences and group discussion; although the quotations mostly describe the patients’ anxiety and psychological distress, the needs still were seen in the description behind the surface of the actual talking. Patients didn’t express their needs directly during the interviews. We added more information to describe the phenomena of whole scenarios to connect the patients’ needs. Please see page 5, line 206-208, page 6, line 210-213, line 229-233, line 246-248, page 7, line 276-278, page 8, line 324-326, line 353-358, and page 9, line 399-405.

  1. Examples Need for companionship- He bent down to pick up the box: “Let's take a good look at her (laughing). Isn’t she beautiful?” He happily placed the box in his shirt pocket- This quite does not reflect need for companionship

In terms of the company need from the interviewer, participant 03 said: “...Walking like this, I mean with someone besides me, makes me feel much better!” During the walk, the old man noticed on the ground a betel nut box showing a picture of a girl in bikini. He bent down to pick up the box: “Let's take a good look at her (laughing). Isn’t she beautiful?” He happily placed the box in his shirt pocket.

 Ans: Thanks for pointing out the irrelevant of the quotation. We added more information to describe the phenomena of whole scenarios to connect the patients’ needs. We also deleted and reworded the quotations for better understanding. Please see page 6, line 238-239.

  1. Need someone to listen- Participant 01 proudly told the interviewer: “I know how to do my laundry using a washing machine…I kept the machine running from evening to dawn. I took out the clothes to dry when I saw it in the morning.” 

 Ans: In this case, Participants were quite upset about not being able to do things that were once they are good at, they hoped there may be someone to listen to them about the limitation and worry of not being able to complete proficient activities anymore. Participants were quite upset about not being able to do things that were once their pride. They need someone to listen to their frustrations about not being able to perform familiar activities. In spite of being diagnosed with moderate dementia, the participants appeared to perceive no severe functional decline and tried their best to not rely on others. However, gradually losing dependence kept them away from engaging in activities they usually performed by their own. For examples, kitchen was no longer their territory, and riding or driving became a totally restricted task. Sometimes, the participants mentioned some tasks they still can do excitedly. We reworded the quotations and provided more description for better understanding. Please see page 5-6, line 206-215.

  1. Some of the quotes are from the carers of the patients. SO were these interviews with patients and their carers – authors need to mention this in detail in the methodology 

 Ans: Yes, these interviews were with patients and their caregivers. Thanks for reviewer’s suggestion. We have added more details of the interview context. Please see page 3, line 118-119.

  1. In tabl, some of the subthemes are not needs.

 Ans: Thanks for reviewer’s comments. After discussion, the authors keep the subthemes, but have re-allocated the sub-themes.

  1. Needs extensive English language editing

 Ans: Thanks for reviewer’s suggestion. the author agree that this is the weakness of our paper. In this revision, we have the manuscript been edited by English language professor. We marked the red words for the changing. Please see the red words for whole paper.

Reviewer 2 Report

Dear Authors,

You have presented paper about dementia, but I have some comments and suggestions about it.

- The abstract should be a single paragraph and should follow the style of structured abstracts, but without headings, due to Instruction for Authors - please remove the headings. 

- In Introduction you have presented „Chinese version of CDR” scale and interpretation of this scale. It is not clear if you have use it as a research instrument or your research base on interview? If this scale was the research tool this information should be in section Materials and Methods. But if not, it seems unfounded to leave this description in the introduction. 

- In section Material and Method, you have indicated that the study was conducted on the basis of an interview, and the results indicate the value of the CDR scale. Please clearly define what was the research tool at work.

- Discussion is quite short and need to extent of other research results.

- In Conclusions there is no practical implications of your research results, add it.

Author Response

To the Reviewer:

Thank you very much for taking time from your busy schedule to comment on our manuscript. Our responses to your suggestions and questions are summarized below, with reference to the appropriate pages in the text. Revisions in the text have also been highlighted in RED. We truly appreciate your thoughtful and constructive comments that help make this a better paper. We are very grateful to you for all the comments and insightful suggestions that help enhance both the quality and readability of our paper. Thank you.

  1. Dear Authors, You have presented paper about dementia, but I have some comments and suggestions about it. The abstract should be a single paragraph and should follow the style of structured abstracts, but without headings, due to Instruction for Authors - please remove the headings. 

 Ans: Thanks for reviewer’s suggestions on the abstract. The authors agree have corrected the abstract. Please see abstract in page 1, line 21-35.

  1. - In Introduction you have presented „Chinese version of CDR” scale and interpretation of this scale. It is not clear if you have use it as a research instrument or your research base on interview? If this scale was the research tool this information should be in section Materials and Methods. But if not, it seems unfounded to leave this description in the introduction. 

 Ans: Thanks for reviewer’s comment. Dementia is a progressive and neurodegenerative disease. It showed big differences in patients with mild and moderate dementia. Patients with mild and moderate dementia have different symptoms and cognitive impairment. In clinical settings, two scales are widely used to identify patients with mild or moderate dementia. Since our study focused on patients with moderate dementia, we need to be pay attention on the recruiting criteria. The Chinese version of CDR was used to screen if the potential participants fit the inclusion criteria or not, and for diagnosing the stage of dementia.

  1. - In section Material and Method, you have indicated that the study was conducted on the basis of an interview, and the results indicate the value of the CDR scale. Please clearly define what was the research tool at work.

 Ans: Thanks for reviewer’s comment. The CDR was used for selecting our participants’ criteria from the clinical settings if the participants met the criteria of moderate dementia. The CDR was not use as a research tool in this study. The authors edited some text in this revision and hop this show clearer for the readers. Thank you for bringing up this issue. Please see page 3, line 106-111.

  1. - Discussion is quite short and need to extent of other research results.

 Ans: Thank you for reviewer’ comment. The authors spent a great deal of effort revising the Discussion section in this revised paper. We really appreciate your insightful analysis and suggestions for our paper. The discussion has been changed a lot. A total of 6 paragraphs was included and hope these changes provide more deep discussion. Please see page 10-12, line 458-539.

  1. - In Conclusions there is no practical implications of your research results, add it.

 Ans: Thanks for reviewer’s suggestion. We added some practical implications in the conclusion. Please see page 12, line 545-550.

Reviewer 3 Report

The study by Yeh et al., focuses on needs of the home-cared people with moderate-stage dementia. The authors emphasize on very relevant and global issue. Overall, the manuscript is well written and exhaustively covers the important topics in the interview.

However, authors mentions that there were seven participants in the interview but the data and experiences of all the participants for all the topics and themes were not included in the manuscript. It is important to mention the experiences of all the participants to justify the purpose of the study.  

In material and method section authors state the inclusion criteria as dementia diagnosed by psychiatrist or neurologist, but they did not mention whether participants have other underlying disease conditions that could contribute to such experience? It would be better if authors could mention the exclusion criteria.

The experiences of individuals are complex and constantly changing that enhances the complexity and makes it even more challenging to capture the soul of dementia experience. The authors should acknowledge the limitation of their study.

Author Response

To the Reviewer:

Thank you very much for taking time from your busy schedule to comment on our manuscript. Our responses to your suggestions and questions are summarized below, with reference to the appropriate pages in the text. Revisions in the text have also been highlighted in RED. We truly appreciate your thoughtful and constructive comments that help make this a better paper. We are very grateful to you for all the comments and insightful suggestions that help enhance both the quality and readability of our paper. Thank you.

  1. The study by Yeh et al., focuses on needs of the home-cared people with moderate-stage dementia. The authors emphasize on very relevant and global issue. Overall, the manuscript is well written and exhaustively covers the important topics in the interview.

 Ans: Thank for reviewer’s positive feedback and for your appreciation of this paper’s value. it really encourages us to carry on our field of research. We really appreciate your insightful analysis and suggestions for our paper.

  1. However, authors mentions that there were seven participants in the interview but the data and experiences of all the participants for all the topics and themes were not included in the manuscript. It is important to mention the experiences of all the participants to justify the purpose of the study.  

 Ans: Thanks for reviewer’s suggestions. However, because of limited space for the paper, the authors cannot provide all 7 participants’ experiences in each subthemes. So we keep the most important and significant data and try equal the representative cases. The following table (only in the response letter) shows the average representative data.

Participants

Theme 1

Theme 2

Theme 3

Theme 4

Total

01

2

1

2

1

6

02

1

1

2

1

5

03

1

2

1

1

5

04

2

1

2

2

7

05

1

1

1

1

4

06

1

1

1

1

4

07

1

2

1

1

5

  1. In material and method section authors state the inclusion criteria as dementia diagnosed by psychiatrist or neurologist, but they did not mention whether participants have other underlying disease conditions that could contribute to such experience? It would be better if authors could mention the exclusion criteria.

 Ans: Thanks for reviewer’s suggestion. We added two exclusion criteria. The exclusion criteria were (1) Premorbid IQ < 70; (2) Current abuse of alcohol or drug, or more than a 5-year history of alcohol or drug abuse. Meanwhile, we added criteria if we need to stop the interviews. The interview was terminated when: 1) became agitated or other unexpected responses during interview, 2) families or home visit nurses though the interview should stop. However, all patients are age more than 65. In addition to dementia, most of study participants have other diseases. We added this information in the new table I in the revised paper. Please see the page 3, line 109-112, and page 3 table 1.

  1. The experiences of individuals are complex and constantly changing that enhances the complexity and makes it even more challenging to capture the soul of dementia experience. The authors should acknowledge the limitation of their study.

 Ans: Thanks for reviewer’s reminding. The authors agree and added the challenging situations happened during data collection, and it did increase the complexity and difficulties in data collection and interpretation. We added this limitation in discussion section. Please see the page 12, line 529-535. Thanks reviewer again. We believe that your suggestions and comments make this paper better.
